# New Insights into the Microstructural Changes During the Processing of Dual-Phase Steels from Multiresolution Spherical Indentation Stress–Strain Protocols

Ali Khosravani [1], Charles M. Caliendo [2] and Surya R. Kalidindi [1,2,*] 

[1] The George W. Woodruff School of Mechanical Engineering, Georgia Institute of Technology, Atlanta, GA 30332, USA; alikhosravani@gatech.edu

[2] School of Materials Science and Engineering, Georgia Institute of Technology, Atlanta, GA 30332, USA; ccaliendo@gatech.edu

[*] Correspondence: surya.kalidindi@me.gatech.edu; Tel.: +1-404-385-2886

**Abstract:** In this study, recently established multiresolution spherical indentation stress–strain protocols have been employed to derive new insights into the microstructural changes that occur during the processing of dual-phase (DP) steels. This is accomplished by utilizing indenter tips of different radii such that the mechanical responses can be evaluated both at the macroscale (reflecting the bulk properties of the sample) and at the microscale (reflecting the properties of the constituent phases). More specifically, nine different thermo-mechanical processing conditions involving different combinations of intercritical annealing temperatures and bake hardening after different amounts of cold work were studied. In addition to demonstrating the tremendous benefits of the indentation protocols for evaluating the variations within each sample and between the samples at different material length scales in a high throughput manner, the measurements provided several new insights into the microstructural changes occurring in the alloys during their processing. In particular, the indentation measurements indicated that the strength of the martensite phase reduces by about 37% when quenched from 810 °C compared to being quenched from 750 °C, while the strength of the ferrite phase remains about the same. In addition, during the 10% thickness reduction and bake hardening steps, the strength of the martensite phase shows a small decrease due to tempering, while the strength of the ferrite increases by about 50% by static aging.

**Keywords:** dual-phase steels; spherical indentation; multi length-scale mechanical testing

## 1. Introduction

Dual-phase (DP) steels with a combination of high tensile strength, high work-hardening rate, and good ductility are being evaluated for the lightweighting of critical structural components in automobiles [1–5]. The desired combinations of properties are achieved in DP steels through the use of multiple thermo-mechanical processing steps. These processing steps typically include intercritical annealing at 730–830 °C for a few minutes up to an hour, quenching at different cooling rates, cold working to different deformation levels, and aging at 100–250 °C up to few hours. The last two steps are typically referred to as bake hardening [6–23]. During the intercritical annealing treatment, the material is heated up to a temperature where the austenite and the ferrite phases are stable. During the subsequent quenching to the room temperature, the austenite transforms to the much harder martensite phase [17,23–29], which essentially controls the properties of the DP steel. It is evident from the Fe–C phase diagram (see Figure 1) that the different intercritical annealing temperatures

will result in different volume fractions of the martensite phase [24]. Furthermore, the amount of carbon content in the martensite (as a solid solution) varies significantly with the chosen intercritical annealing temperature (for example, it can change from 0.17 to 0.77 wt.% when the intercritical annealing temperature is reduced from 830 and 730 °C), while the corresponding change in the ferrite phase is insignificant (only about 0.01–0.02 wt.%). Furthermore, the relevant section of the phase diagram (see Figure 1) suggests that intercritical annealing at lower temperatures results in a smaller volume fraction of martensite but with a higher carbon content. As already mentioned, another important component in the processing of DP steels is the bake hardening (BH) [6–23] step, which includes cold working followed by aging heat treatment. This step is known to impart DP steels with a characteristic property known as continuous yielding, which is generally attributed to the production and pinning of dislocations in the ferrite component, especially in the vicinity of ferrite/martensite interfaces [1,7–14,16,18,19,22,28,30–34].

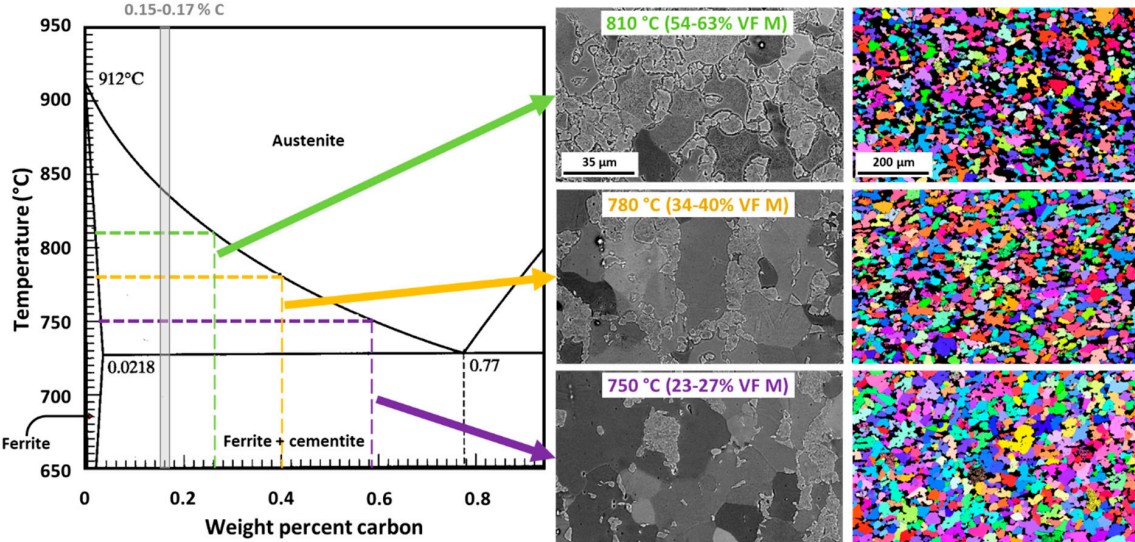

**Figure 1.** Small section of the binary Fe–C phase diagram showing the different compositions and microstructures produced from three different intercritical annealing treatments on the initial low carbon steel with 0.15–0.17 wt.% carbon content. The intercritical annealing temperatures selected for this study were 750 °C, 780 °C, and 810 °C. The microstructures are shown as back-scattered electrons (BSE) and electron-backscattered diffraction (EBSD) maps.

The discussion above points out the difficulties encountered in the optimization of the processing of DP steels to meet the desired combination of properties [1,35,36]. This is mainly because of the need to explore a very large and complex process space (each element of this space should specify the entire process history, including the complete sequence of substeps described earlier). One of the central bottlenecks comes from the lack of reliable information on the changes in the properties of the constituent phases (i.e., martensite and ferrite) as a function of the process parameters. A number of prior studies [37–52] have employed indentation techniques for this task. However, most of these studies have employed sharp indenters and reported large variances in the measured values. A summary of such measurements in DP steels is presented in Table 1, which shows a range of 2–7 GPa for the hardness of the ferrite phase and a range of 3–13 GPa for the hardness of the martensite phase. The large variations in the reported hardness data have hindered attempts aimed at extracting quantitative physical insights that could guide the rational design of DP process histories to achieve desirable combinations of bulk properties. In recent work [53–61], it has been demonstrated that indentation yield strength is a much more reproducible and reliable measure of the intrinsic plastic strength of the microscale constituents in a heterogeneous material, and could be estimated from the recently established spherical indentation stress–strain protocols [62,63]. Using this analysis method, it has been shown [57] that the indentation

yield strength is very sensitive to the carbon content in the lath martensite; increasing C content from 0.13 to 0.30 wt.% improves the indentation yield strength by 42–48% and the indentation work hardening by 27–47%. In this work, we extend and employ these techniques to provide quantitative insights into the changes in the yield strengths of the martensite and ferrite phases in DP steels, especially during the intercritical annealing and the bake hardening steps.

**Table 1.** Summary of hardness measurements in martensite and ferrite from prior literature.

| Alloy Grade/Composition | Indenter Type | Max Depth (nm) | Max Load (mN) | Martensite Hardness (GPa) | Ferrite Hardness (GPa) | Reference |
|---|---|---|---|---|---|---|
| DP780 | Berkovich | - | 5 | 6.2 ± 0.11 | 3.7 ± 0.03 | 37 |
| DP1300 | Berkovich | - | 5 | 4.5-10 | 2-5 | 38 |
| 0.1C5Mn3Al | Berkovich | - | 5 | 4.7 ± 0.4 | 4.1 ± 0.3 | 39 |
| 0.1C5Mn3Al (60% TR) | Berkovich | - | 5 | 5.9 ± 0.7 | 5.0 ± 0.5 | |
| DP980 | Berkovich | - | 2.5 | 3-10 | 1.5–5.5 | 40 |
| DP980 (7% strain) | Berkovich | - | 2.5 | 3-13 | 2–7 | |
| (0.04, 0.07, 0.1) C1.2Mn0.15Si | spherical (R = 2.8, 5.7 μm) | - | 15 | - | 1.8–2, 1.3–1.6 | 41 |
| DP980 | Berkovich | 40 | - | 4.5–9 | 3–4.75 | 42 |
| API-X100 | Spherical (R = 0.5, 3 μm) | - | 15, 30 | - | 3.4–4.1, 1.9–2.4 | 43 |
| 0.16C1.5Mn1Si | cube-corner | - | 1 | 6.3–7.9 | 2.8 | 44 |
| 0.19C1.6Mn0.2Si | Berkovich | - | - | 3–10.8 | 2.8-6.8 | 45 |
| 0.18C0.75Mn0.4Si | Berkovich | - | 10 | 7.6 | 2.2 ± 0.2 | 46 |
| 0.38C0.67Mn0.2Si | Berkovich | - | 10 | 4.9–7.3 | - | 47 |
| DP980 | Berkovich | 50 | - | 6–11 | 4–5.5 | 48 |
| DP980 | Berkovich | - | 0.8 | 8.4 ± 0.9 | 4.1 ± 0.3 | 49 |
| 0.08C1.74Mn0.75Si | Berkovich | - | 0.05 | 4.5–5.5 | 3–3.5 | 50 |
| DP980 | Berkovich | - | 3 | 6.3–8.1 | 2.5–3.5 | 51 |
| DP590 | Berkovich | 50 | - | 3.5–4.1 | 1.5–1.8 | 52 |

## 2. Materials and Method

### 2.1. Sample Preparation

A four mm thick strip of low carbon steel with a chemical composition (in wt.%) of 0.16C, 1.4Mn, 0.04P, and 0.04S was used to produce the DP steel samples needed for this study. Small coupons of the low carbon steel with dimensions of 10 mm × 20 mm × 4 mm were cut, and heat-treated at 450 °C for 2 h to obtain a starting annealed microstructure. Three intercritical annealing temperatures of 750 °C, 780 °C, and 810 °C were selected to produce different volume fractions of the martensite phase after quenching. The heat treatment was carried out in a molten salt bath, LIQUID HEAT 168 from Houghton International (Houghton International Inc., Norristown, PA, USA), to ensure quick heating and uniform temperature inside the small coupons. After holding three samples for 4 min at each selected intercritical annealing temperature, they were quenched in an oil bath to room temperature. Of each set of three samples thus produced, one sample was retained without bake hardening. The other two samples from each set were subjected to thickness reductions of 5% and 10% by cold rolling, respectively, and heat-treated at 170 °C for 20 min and quenched in water. As a result of the protocols described above, a total of nine samples with nine different processing conditions were produced. The sample labeling was designed to reflect the processing history in the form "intercritical annealing temperature-thickness reduction percentage-bake hardening temperature". As an example, sample 810-10-170 indicates quenching after an intercritical annealing temperature of 810 °C followed

by 10% thickness reduction and bake hardening at 170 °C for 20 min. Likewise, sample 810-00-000 indicates that the sample was subjected to only intercritical annealing at 810 °C followed by quenching.

Samples were prepared for microscopy and indentation using standard metallography procedures. This included grinding with silicon carbide papers down to a grade of 4000 and polishing sequentially with suspensions of 3 µm and 1 µm diamond particles. The final step of polishing included vibro-polishing (Struers Inc., Cleveland, OH, USA) for 24 h using colloidal silica suspension. After polishing, SEM (scanning electron microscopy) images and EBSD (electron backscatter diffraction) (EDAX Inc., Mahwah, NJ, USA) maps were obtained from all samples using a TESCAN MIRA3 (TESCAN USA Inc., Warrendale, PA, USA) scanning electron microscope with a field emission gun set at 20 kV. High contrast in electron channeling contrast image (ECCI) at sub-micron resolution was achieved with a working distance of 5–6 mm and a voltage of 30 kV.

### 2.2. Spherical Nano-Indentation Stress–Strain Protocols

After imaging, spherical nanoindentation tests were carried out in an Agilent G200 Nanoindenter (KLA Inc., Milpitas, CA, USA). As these measurements were aimed at obtaining responses from the ferrite and martensite regions in the sample, smaller indenter tips of radii 1 µm and 16 µm were employed in these tests. At least 20–30 indentation measurements were conducted for each phase (i.e., ferrite and martensite) on each sample. A constant strain rate of 0.05 s$^{-1}$ was used with a maximum depth of 200 nm and 350 nm for the indenter tips with radii of 1 µm and 16 µm, respectively. The Agilent G200 Nanoindenter used in this study had an XP head and CSM (continuous stiffness measurement) module. The CSM superimposes small sinusoidal load/unload cycles on the monotonic loading history with a frequency of 45 Hz and an amplitude of 2 nm. The CSM capability allows an accurate estimation of contact radius used in calculating indentation stress and indentation strain [62,63]. The analysis protocols used in the nanoindentation tests are briefly presented next.

Let $P$, $h_e$, $E_{eff}$, and $R_{eff}$ denote the indentation load, the elastic indentation depth, the effective modulus of the indenter-sample system, and the effective radius of the indenter-sample system, respectively (see Figure 2). These variables can be related to each other using Hertz's theory [64] for elastic contact between two isotropic bodies as

$$P = \frac{4}{3} E_{eff} \sqrt{R_{eff} h_e^3},  \tag{1}$$

$$\frac{1}{E_{eff}} = \frac{1 - v_i^2}{E_i} + \frac{1 - v_s^2}{E_s},  \tag{2}$$

$$\frac{1}{R_{eff}} = \frac{1}{R_i} + \frac{1}{R_s},  \tag{3}$$

where $E_i$, $v_i$ and $E_s$, $v_s$ are Young's modulus and Poisson ratio for the indenter and the sample, respectively. For the initial elastic loading when the surface of the sample is still flat, the effective radius, $R_{eff}$, is equal to the indenter radius, $R_i$. Therefore, it would be possible to extract $E_{eff}$ from the measured load–displacement using standard regression techniques. One of the central challenges in this analysis comes from the need for a highly accurate estimation of the effective point of the initial contact (i.e., zero-point correction) [62,63,65–71]. This zero-point correction helps in dealing with many of the unavoidable issues encountered at initial contact, including imperfections in indenter shape and non-ideal surface conditions (e.g., oxide layer, surface roughness). In the protocols used in this work, the initial contact point was determined by finding load and displacement corrections ($P^*$ and $h^*$) from the following relation derived from Equation (1) [62]:

In Equation (4), $S$ is the measured elastic unloading stiffness obtained using the CSM capability mentioned earlier, and $\widetilde{P}$ and $\widetilde{h}$ are the raw measurements of load and displacement, respectively. To estimate $P^*$ and $h^*$, Equation (4) is re-cast as

$$S = \frac{3P}{2h_e} = \frac{3}{2}\frac{(\widetilde{P} - P^*)}{(\widetilde{h} - h^*)}. \tag{4}$$

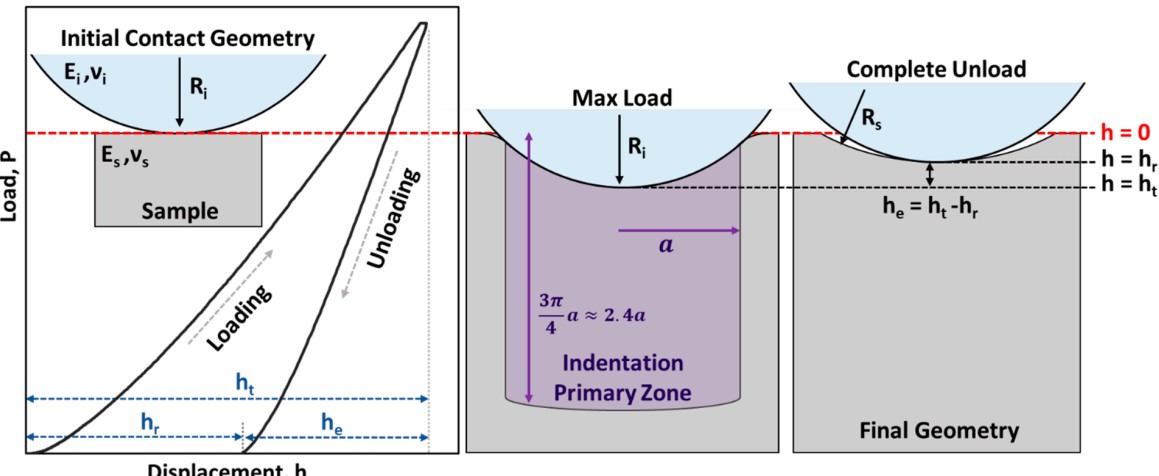

**Figure 2.** Schematic of a typical measured spherical indentation load–displacement curve and their corresponding initial and final contact geometries. The indentation primary zone within which the majority of deformation occurs is highlighted as a purple cylinder with a radius and height of *a* and 2.4 *a*, respectively.

The form of Equation (5) allows an accurate estimation of the zero-point corrections (i.e., values of $P^*$ and $h^*$) by performing linear regression between the measured values of $\widetilde{P} - \frac{2}{3}S\widetilde{h}_e$ and $S$;

$$\widetilde{P} - \frac{2}{3}S\widetilde{h}_e = -\frac{2}{3}h^*S + P^*. \tag{5}$$

Figure 3a–c illustrates the main steps involved in the analyses of the nanoindentation measurement reports in this study. An example measured raw load–displacement data as shown in Figure 3a. The application of the zero-point correction described in Equation (5) is illustrated in Figure 3b, where the expected linear portion based on Hertz's theory is shown in yellow (the corresponding segment in the load–displacement curve is shown in the inset in Figure 3a). After the zero-point correction, $E_{eff}$ can be estimated by performing a linear regression between $P$ and $h^{3/2}$ in the initial elastic portion of the measured load–displacement curve (highlighted in yellow in Figure 3a) [62]. The estimated value of $E_{eff}$ is then used to estimate the continuously evolving contact radius, *a*, using the following relationship derived from Hertz's theory [64]:

$$a = \frac{S}{2E_{eff}}. \tag{6}$$

An important aspect of Equation (6) is that it is applicable at any point in the complex elastic-plastic loading-unloading cycles applied to the sample. By Equation (6), the effective indentation modulus measured from the initial elastic loading segment is assumed to remain constant throughout the elastic–plastic loading applied to the sample. Estimation of the continuously evolving contact radius allows the estimation of indentation stress and indentation strain [62] defined as follows:

$$\sigma_{ind} = \frac{P}{\pi a^2}, \tag{7}$$

$$\varepsilon_{ind} = \frac{4}{3\pi}\frac{h_t}{a}. \tag{8}$$

An example indentation stress–strain curve extracted using the above protocols presented in Figure 3c. The spherical nanoindentation stress–strain protocols described above have been validated extensively in both experiments [53–61,72–95] and numerical simulations (performed using finite element models) [96–99]. As a result of these prior validations, we are now fairly confident in obtaining highly reproducible indentation stress–strain curves on a broad variety of material samples.

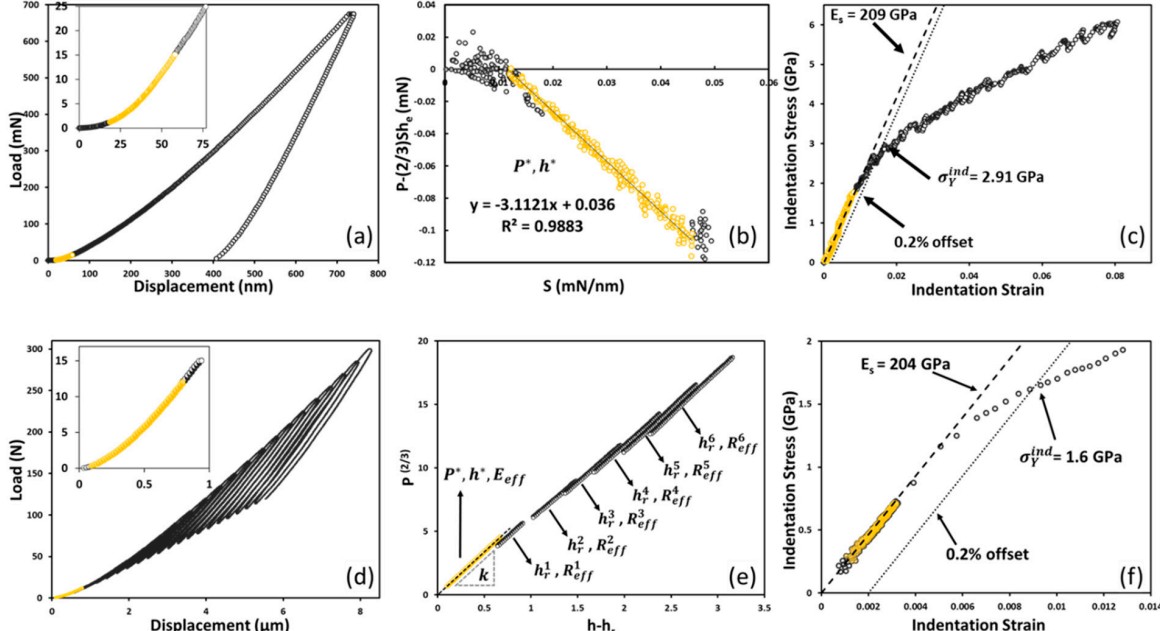

**Figure 3.** (**a**) Typical load–displacement curve from the nanoindentation test and the initial elastic segment (highlighted in yellow) identified in (b). (**b**) The identification of the effective zero-point by linear regression analyses of the linear portion of the curve between $P - \frac{2}{3}Sh_e$ and $S$. This linear regression allows the estimation of the zero-point corrections, $P^*$ and $h^*$. (**c**) Indentation stress–strain curve extracted from the corrected nanoindentation data. (**d**) Typical load–displacement curve from the microindentation test in absence of CSM (continuous stiffness measurement) measurements, using a multitude of unloading segments. The initial elastic segment is highlighted in yellow. (**e**) The identification of the effective zero-point by linear regression analyses of the straight line between $P^{2/3}$ and $h - h_r$. This analysis allows the estimation of $P^*$ and $h^*$. (**f**) Indentation stress–strain curve extracted from the corrected data for the microindentation test.

### 2.3. Spherical Micro-Indentation Stress–Strain Protocols

As already noted earlier, our interest here also includes bulk plastic properties of the various samples produced for the study. The bulk mechanical response of each sample was measured using a customized Zwick-Roell Z2.5 hardness (microindentation) tester (ZwickRoell Group, Ulm, Germany). For these tests, a spherical indenter with a 6350 μm radius indenter tip was used. The indenter is made of tungsten carbide to ensure high rigidity of the indenter. A total of 10–15 tests were conducted at randomly selected locations on each sample. A constant crosshead speed of 0.1 mm/min was used in all microindentation tests reported in this study.

As the Zwick-Roell Z2.5 tester does not have CSM capability, one needs a different strategy to extract indentation stress–strain curves from the spherical microindentation measurements. In recent work, it was shown that one can estimate the contact radius reliably from superimposed unloading segments (corresponding to about 30–50% of the peak force) when the CSM option is not available [63]. Figure 3d shows an example measurement of the load–displacement data in multiple loading-unloading cycles in the spherical microindentation protocols employed in this work. In these protocols, each unloading segment is assumed to be purely elastic and is analyzed using Hertz's theory [64]:

$$\left(\widetilde{h}_e - h^*\right) = k\left(\widetilde{P} - P^*\right)^{\frac{2}{3}}, \tag{9}$$

$$k = \left[\frac{3}{4}\frac{1}{E_{eff}}\frac{1}{\sqrt{R_{eff}}}\right], \tag{10}$$

where $P^*$ and $h^*$ represent once again the zero-point corrections. To estimate the zero-point load and displacement corrections, $P^*$ and $h^*$, a least-squares regression was applied to the initial measured load–displacement data in the elastic regime in Equation (9). As the indentation starts on a flat surface, $R_{eff}$ in the initial elastic regime (highlighted in yellow in Figure 3d) is equal to the indenter radius, $R_i$. Thus, one can estimate $E_{eff}$ from Equation (10), which is assumed to remain constant even after the specimen has undergone plastic deformation.

Plastic deformation under the indenter leads to a continuous evolution of $R_{eff}$. Each subsequent unloading (after the estimation of $E_{eff}$ from the initial elastic loading segment) is analyzed using Hertz's theory, but this time with a focus on estimating the evolving values of $R_{eff}$ and the values of the indentation contact radius, $a$. This is accomplished by regressing each unloading curve in Figure 3d to Hertz's theory expressed as

$$h - h_r = k(P)^{\frac{2}{3}}, \tag{11}$$

where $h_r$ denotes residual displacement (after complete unloading; see Figure 2). This regression analysis allows the determination of the values of $h_r$ and $R_{eff}$ (see Equation (10)) corresponding to each unloading segment obtained in the test. Figure 3e illustrates the above protocol for six selected unloading segments (out of a much larger number of unloading segments depicted in Figure 3d). The values of the contact radius $a$ are then estimated using Hertz's theory as

$$a = \sqrt{R_{eff}(h_{s,max} - h_r)}, \tag{12}$$

where $h_{s,max}$ is the indentation displacement in the sample at the peak of each unload. Once the contact radius is estimated, the values of indentation stress and indentation strain can be computed using Equations (7) and (8). It should be noted that each unloading segment in these protocols results in the estimation of one point on the microindentation stress–strain curve. Consequently, multiple load–unloading cycles are needed to produce a reasonable indentation stress–strain curve. Twenty unloading segments were incorporated in each microindentation test reported in this study. An example microindentation stress–strain curve extracted in our study is shown in Figure 3f.

## 3. Results and Discussion

Example SEM and EBSD images obtained from the DP steel samples produced in this work are presented in Figure 1. SEM images reveal the presence of martensite islands in a matrix of ferrite grains. For the selected low carbon alloy with carbon content in the range of 0.15–0.17 wt.%, samples quenched from 750 °C, 780 °C, and 810 °C were expected to produce 23–27 vol.%, 35–40 vol.%, and 56–64 vol.% of martensite, respectively. These predicted values were confirmed from several large EBSD scans obtained at different locations on each sample (presented in Table 2). In the EBSD maps, multicolored regions are ferrite grains whose lattice orientations have been mapped out with a 1 μm spatial resolution and martensite regions are colored black based on low IQ (image quality) compared to the ferrite regions. As seen from the phase diagram in Figure 1, quenching from a higher intercritical annealing temperature will result in a higher volume fraction of lower carbon content martensite in the microstructure. Some prior studies have shown the significant effect of carbon content on the hardness of the martensite by systematically changing C content in the tested materials [100,101]. As already mentioned, the protocols used in prior literature using sharp indenters result in high variances between different studies (see Table 1). In the present work, we employed the spherical indentation stress–strain

protocols that have been shown to produce reliable and highly reproducible values for indentation yield strengths at multiple material length scales.

**Table 2.** Summary of the microindentation measurements obtained in this study. The martensite volume fractions were estimated from large EBSD scans collected on each sample.

| Sample Code | Martensite Volume Fraction (%) | Average Elastic Modulus (GPa) | Average Indentation Yield Strength (MPa) | Contact Area Diameter at Yield Point (μm) |
|---|---|---|---|---|
| 750-00-000 | 25.4 | 174.5 ± 25.3 | 899.5 ± 62.9 | 284.6 ± 14.0 |
| 750-05-170 | 27.5 | 193.4 ± 23.3 | 950.4 ± 29.7 | 333.4 ± 26.4 |
| 750-10-170 | 23.5 | 200.9 ± 22.3 | 1100.9 ± 130.3 | 352.4 ± 23.0 |
| 780-00-000 | 34.8 | 216.8 ± 23.3 | 1097.7 ± 41.6 | 331.4 ± 29.6 |
| 780-05-170 | 35.3 | 209.6 ± 15.4 | 1299.4 ± 68.7 | 356.2 ± 35.4 |
| 780-10-170 | 38.9 | 201.9 ± 16.7 | 1336.3 ± 69.5 | 362.6 ± 28.4 |
| 810-00-000 | 56.3 | 188.5 ± 14.7 | 1168.6 ± 186.8 | 306.6 ± 21.8 |
| 810-05-170 | 59.6 | 219.2 ± 14.5 | 1340.4 ± 130.9 | 338.6 ± 20.4 |
| 810-10-170 | 60.4 | 207.6 ± 9.7 | 1506.5 ± 132.3 | *361.0 ± 30.2* |

*3.1. Microindentation and Results*

About 10–15 microindentation tests were conducted at randomly selected locations on each of the nine differently processed samples. From each microindentation test, values of Young's modulus and indentation yield strength (defined using a 0.2% plastic strain offset as shown in Figure 3f) were extracted; these are summarized in Figure 4 and Table 2. It is seen from Table 2 that increasing the intercritical annealing temperature increased the bulk indentation yield strength of the sample (e.g., the indentation yield strength increased by ~30%, from 899.5 ± 62.9 MPa for sample 750-00-000 to 1168.6 ± 186.8 MPa for sample 810-00-000). In addition, the same trend can be detected when cold work and bake hardening processes are applied. The indentation yield strength increased by 17–29%, when 10% thickness reduction and bake hardening were applied, when compared to the quenched sample without bake hardening. As a specific example, the indentation yield strength increased from 1168.6 ± 186.8 MPa for sample 810-00-000 to 1506.5 ± 109.5 MPa for sample 810-10-170.

In the indentation tests, the contact diameter ($2a$) provides an estimate of the length scale of the material under the indenter that has been subjected to significant plastic deformation (see Figure 2). The estimated contact diameters at the indentation yield estimated by Hertz's theory (Equation (12)) are summarized in Table 2. As an example, the estimated contact diameter at the yield point for one of the microindentations conducted in this study is shown as a blue dashed circle on a sample micrograph in Figure 4d. Furthermore, the primary deformation zone under the indenter is estimated to extend to about 2.4$a$ underneath the indenter [62]. It is therefore estimated that the indentation zone at yield is approximately 300 μm in diameter and 400 μm in depth for the samples studied here. The primary indentation zone at yield in the microindentation tests reported in this study contained ~300–700 grains (based on an approximate grain size of 30 μm). Therefore, the microindentation measurements presented here can be assumed to reflect the bulk (macroscale) properties of the samples.

The microindentation measurements presented here compare well with the values reported in the literature for samples with similar compositions and processing histories. The reported values of Young's moduli for similar DP steels are in the range of 170–210 GPa [102–104]. Furthermore, tensile yield strengths have been reported in the range of 345–482 MPa [24,105–107], 413–622 MPa [24,106,108,109], and 520–670 MPa [110–113] for the intercritical annealing temperatures of 750 °C, 780 °C, and 810 °C, respectively. Considering a scaling factor of 2 (see Patel et al. [97]) for converting indentation yield

strength to the tensile yield strength, the values reported in Table 2 are in good agreement with the values reported in the literature.

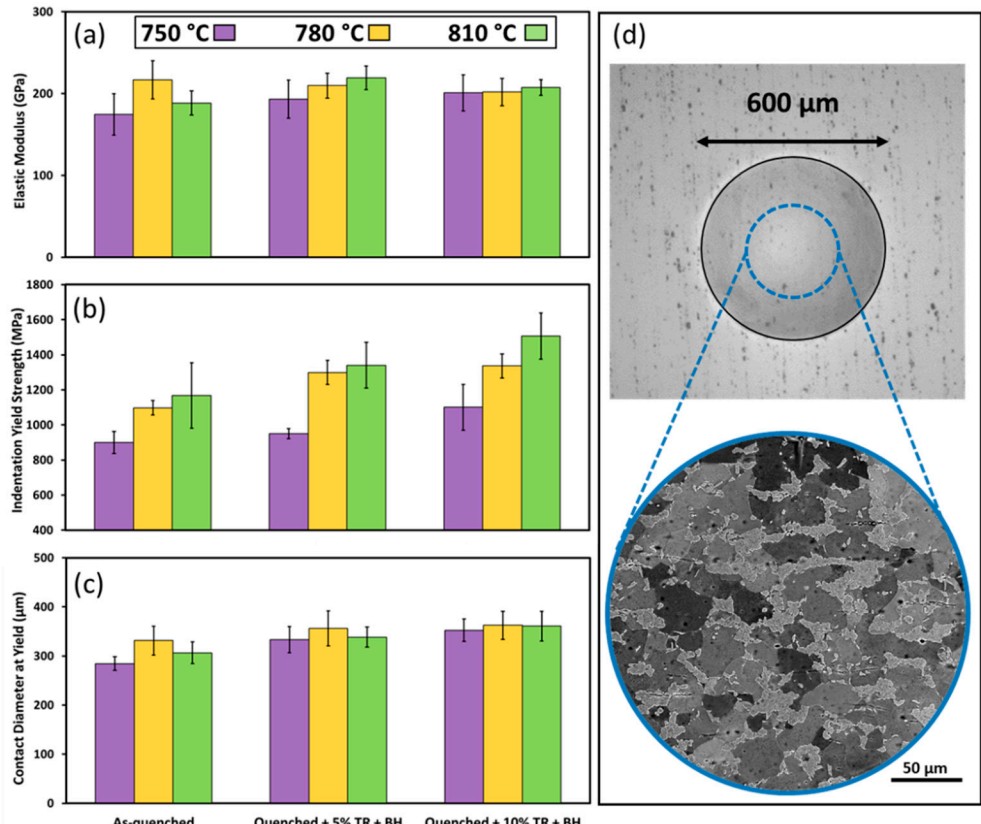

**Figure 4.** (**a–c**) Histograms summarizing the extracted values of Young's moduli, indentation yield strengths, and the contact diameters at yield from microindentation tests on samples produced by different processing histories. The different processing histories included intercritical annealing at 750 °C, 780 °C, and 810 °C followed by quenching. The samples were then cold-worked (5% and 10%) and bake hardened (BH). (**d**) An optical micrograph and BSE image showing the indentation size at yield as a dashed blue circle.

The results from the microindentation measurements raise two important questions. First, when the martensite volume fraction doubled (from ~25% to ~58%), the indentation yield strength increased only by ~30%. Given the expected high strength of the martensite compared to the ferrite, one should expect a bigger increase in the macroscale yield strength. Second, the bake hardening produced an increase in the strength by 1729%. Given that the martensite volume fraction remains the same during the bake hardening process, and a possible reduction of the martensite strength during the bake hardening step, the physical processes responsible for this significant increase in the strength are not clear. These questions can be addressed by measuring the mechanical responses at the length scale of the constituent phases using the nanoindentation protocols described earlier.

*3.2. Nanoindentation and Results*

As discussed earlier, DP steel microstructures exhibit rich heterogeneity over a hierarchy of material length scales (see Figure 5). At the mesoscale, these include different thermodynamic phases with different crystal structures (Figure 5a,b) and local differences in chemical compositions (especially in carbon and alloying elements). At lower length scales, there exist other heterogeneities within each phase. In martensite, one encounters grains of different crystal orientations, low and high angle grain/phase boundaries, and possibly small carbides (Figure 5c,d). In ferrite, one observes heavily

deformed ferrite regions, especially at the vicinity of the ferrite/martensite interface (Figure 5e). Studies of the mechanical responses of DP steels at different spatial resolutions are critically needed to obtain new physical insights into the overall response of the alloy. Given the heterogeneity involved, it is also important to perform multiple measurements at randomly selected locations to obtain statistically meaningful data.

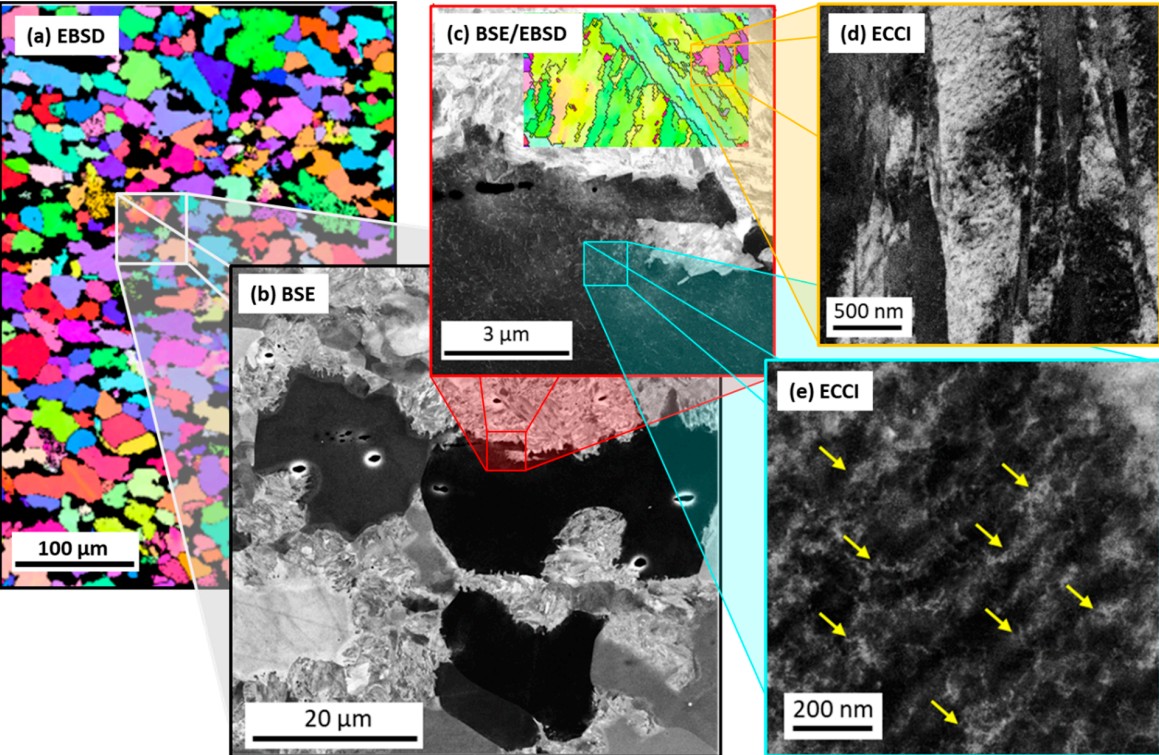

**Figure 5.** A sampling of microstructures of the sample 810-00-000 using different techniques showing the heterogeneous and hierarchical microstructure of DP steel at different length scale. (**a**) An EBSD map showing several ferrite grains (in different colors) and martensite regions (in black color). (**b**) A BSE-SEM image of the microstructure at slightly higher magnification shows details of the martensite regions that contain blocks of martensite reflected with high contrast inside each island. (**c**) The magnified image shows more detail at the martensite/ferrite interface. The colored image is a high-resolution EBSD map on the martensite island showing the orientations of martensite blocks. (**d**,**e**) High-resolution maps of the martensite and ferrite regions obtained using the electron channeling contrast imaging (ECCI) technique. The ECCI micrograph on the martensite shows a highly dislocated structure. The ECCI micrograph on the ferrite grain were collected at the vicinity of the martensite/ferrite interface, where a higher dislocation network (highlighted by yellow arrows) was observed compared to the regions in the center of the ferrite grains.

In this study, we investigated the local mechanical responses in the DP steel samples using two different spherical indenter tips of radii 1 μm and 16 μm, respectively. In prior work on fully martensitic Fe-Ni alloys [57], the contact diameters in indentations conducted with a 1 μm radius tip were estimated to be ~100–150 nm at yield. The primary deformation zone in these indentations is likely to have included only a handful of martensite laths, as their thickness was reported to be in the range of 50–200 nm [114,115]. Furthermore, it was reported [57] that the measured average indentation yield strengths with both the 1 μm and the 16 μm tip radii were in good agreement with each other, while the standard deviations were smaller for the measurements with the 16 μm tip. This is reasonable because one would expect a larger number of martensite laths in the primary indentation zone with the larger indenter tip. For the DP steel samples studied here, it was observed that the martensite indentation yield strengths measured with the 16 μm tip were systematically lower compared to the measurements with

the 1 µm tip. For example, for the case of sample 810-10-170, the measured values of the Young's modulus and indentation yield strength on martensite regions using the 16 µm tip radius were 179.5 ± 10.6 GPa and 2.15 ± 0.26 GPa, respectively. However, the values of Young's modulus and indentation yield strength measured using the 1 µm tip radius were 241.4 ± 20.3 GPa and 3.0 ± 0.46 GPa, respectively. This observation is particularly surprising as neither the fully martensitic alloy [57] nor other metals we have previously tested using similar protocols [57,59–61] showed any strong effects of indenter tip size on the measured indentation yield strengths. We believe that the lower values measured with the 16 µm tip are a consequence of the fact that the stress fields under the indenter extend far beyond the primary indentation deformation zone (defined to be of the order of the contact diameter). Indeed, finite element simulations conducted in prior studies from our research group [62] have revealed that the stress field underneath the indenter can extend as far as 10*a* to 15*a* before reducing to negligible levels. This is also the primary reason why the existing Vickers, Knoop, and Rockwell hardness standards [116,117] recommend that the thickness of the tested sample should be at least 10 times the indentation depth to minimize any influence of the substrate on the measured indentation properties. In the DP steel samples studied here, the martensite particles are surrounded by soft ferrite grains. Given the typical martensite particle size of 10 µm (see Figure 1) the average thickness of the martensite region in the indentation direction is likely to be only ~5 µm (i.e., if one assumes that half the particle has been polished to reveal the martensite region on the sample). Based on the above discussion, we estimate that the stress field under the 16 µm tip in the indentations on martensite regions is likely to extend to about 10 µm, which is significantly larger than the expected remaining thickness of the martensite plate at the indentation site in our measurements. The above discussion is schematically illustrated in Figure 6 where the indentation primary zone (volume of material underneath indentation where the majority of plasticity occurs) is highlighted in red and the extent of the indentation stress field is highlighted in yellow for the 1 µm and 16 µm tip radii on a BSE-SEM image of a typical sample studied in this work. Although the primary indentation zone with the 16 µm tip is expected to lie within the martensite region, the corresponding stress field should be expected to extend into the soft ferrite region underneath the indented martensite particle. For the indentations performed with the 1 µm tip radius, both the primary indentation zone and the indentation stress field are much more likely to lie within the martensite phase. It is also important to note that the local stresses in the primary indentation zone in the martensite are expected to be high because of the higher hardness of the martensite phase. Indeed, the softer ferrite regions at the martensite-ferrite boundary under the 16 µm tip indenter might even experience some plastic deformation. Because of the factors discussed above, we believe that the lower values of Young's moduli and the indentation yield strengths measured on the martensite regions with the 16 µm tip are largely a consequence of the softer ferrite underneath the indented martensite particles. Consequently, only the indentation measurements on martensite regions obtained using the 1 µm tip are reported in this work.

For the nanoindentation on ferrite grains, it was observed that the measurements obtained with the 1 µm tip exhibited high levels of variance within a single sample. We believe this is because of the heterogeneously deformed regions present in the ferrite grains (see Figure 5b,c,e). In particular, higher levels of dislocation density have been noted in the vicinity of the martensite/ferrite interface [1,11,118,119]. To obtain reliable measurements of indentation properties from the ferrite regions, we identified relatively large ferrite grains and used the 16 µm tip. The softer ferrite limits the stress values in the primary indentation zone, further mitigating any influence of the harder martensite regions below the indented ferrite grains.

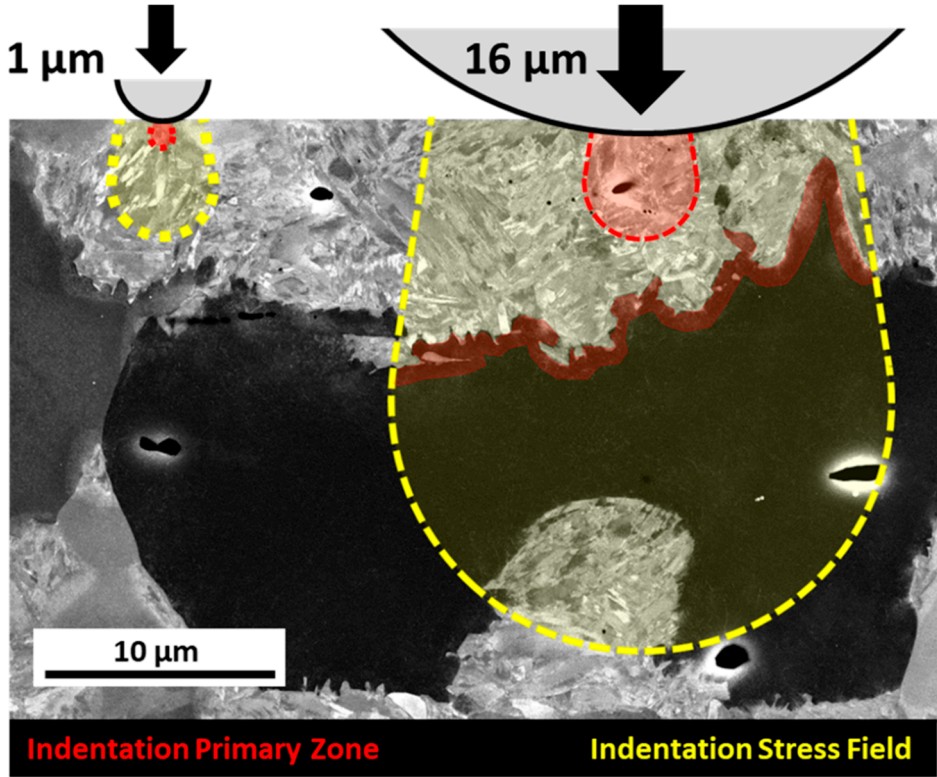

**Figure 6.** Schematic illustration of the primary indentation deformation zone (volume of the material where the majority of deformation occurs) and the extent of the indentation stress field for both the 1 μm and 16 μm tip radii, superimposed on a representative BSE-SEM image of DP steel studied in this work.

Examples of the indentation load–displacement and indentation stress–strain curves obtained in this work are presented in Figure 7 for both the martensite and ferrite regions in sample 750-00-000. As discussed before, the transition from the elastic regime to the elastic-plastic regime is clearly discernable in the indentation plots. The values of Young's modulus for the martensite regions were found to be higher compared to those measured in the ferrite regions (in Figure 7, these values were 226 GPa and 176 GPa, respectively). The indentation yield strength is usually defined using a 0.002 plastic strain offset. This protocol has been used extensively in prior studies [54–59], especially in the absence of pop-in events. A pop-in is identified as a sudden jump in the indentation displacement at a roughly constant load (the tests are performed in load control) and appears as a strain burst in the indentation stress–strain curves (see the measurement for the ferrite region in Figure 7). Pop-in events have been observed extensively in previous work [54,120–125] and were attributed to the difficulty of activating dislocation sources in the very small primary indentation zone. These pop-ins make it difficult to accurately estimate the indentation yield strength. In prior work [54,55,58,61], it was shown that a back-extrapolation method (see Figure 7) provides a reasonable estimate of the indentation yield strength in such cases. This same strategy was employed in the present work.

As discussed earlier, the intercritical annealing temperature and bake hardening steps are the critical processing steps in manufacturing DP steels. The effect of the quenching temperature on the strength of the individual ferrite and martensite regions was studied using samples 750-00-000, 780-00-000, and 810-00-000. As already mentioned, these samples produce martensite regions with significant differences in C content, offering an opportunity to quantitatively study the effect of C content on the yield strength of martensite. Comparing the nanoindentation measurements in the martensite and ferrite regions in samples 810-00-000 and 810-10-170 will provide quantitative insights into how bake hardening influences the yield strengths of these constituents. About 25–30

nanoindentation tests were performed in centers of randomly selected, relatively large, ferrite, and martensite regions on each sample identified above.

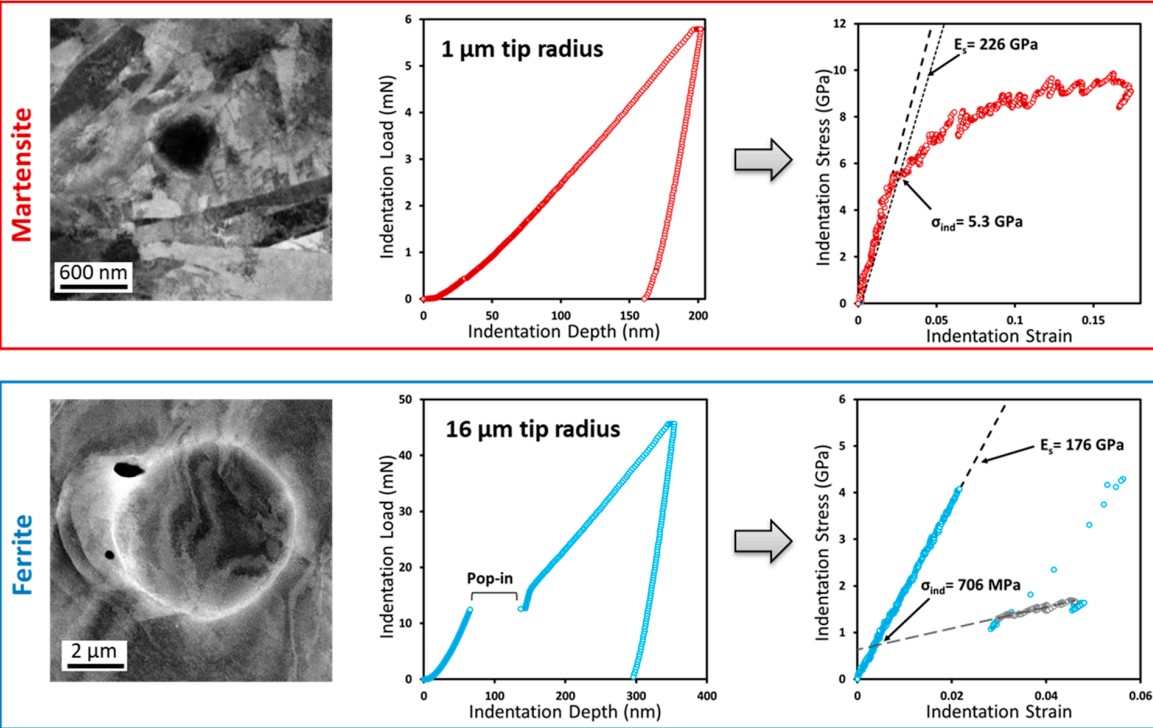

**Figure 7.** Example load–displacement, indentation stress–strain curves, and SEM micrographs of residual indentations from the martensite (top in red color) and ferrite (bottom in blue color) regions.

Figure 8 summarizes all of the indentation stress–strain curves extracted from the tests conducted on the four samples identified above. For reasons already discussed, the 1 μm tip was used on martensite regions, the 16 μm tip was used on ferrite regions, and the 6.35 mm tip (microindentation) was used for evaluating the bulk properties of the samples. The Young's moduli for martensite and ferrite regions were found to be significantly different from each other in these samples. The Young's modulus in the martensite regions varied in the 220–252 GPa range, while it varied in a much narrower range of 171–179 GPa for the ferrite regions. The larger variance in the extracted values of Young's moduli for the martensite regions can be attributed to the fact that the smallest tip (1 μm radius) was used in these studies. The limited number of laths in the indentation zone combined with the expected significant effect of the martensite lattice orientation on the indentation measurements can explain the observed larger variance. Notwithstanding the larger variance, the measurements clearly indicate that the Young's modulus of the martensite is higher compared to the Young's modulus of the ferrite.

The average indentation yield strength (± one standard deviation) is shown as a colored band on each plot in Figure 8. The indentation yield strengths measured from all four samples mentioned earlier are summarized in Figure 9. As expected, the measured variances decrease with an increasing indenter tip radius. This is because the larger indenter tips provide averaged responses over larger material volumes. Figure 9 also presents percentage changes in the indentation yield strengths measured in the martensite and ferrite regions in samples 780-00-000 and 810-00-000 using sample 750-00-000 as the baseline. It is seen that the bulk indentation yield strength increased by 27% when the intercritical annealing temperature was increased from 750 °C to 810 °C. As already noted, there is a significant increase in the martensite volume fraction (25% for 750-00-000 samples and 56% for 810-00-000 samples; see Table 2). Figure 9 also shows that the martensite yield strength has decreased by ~37% in the 810-00-000 sample, compared to the 750-00-000 sample. This reduction is attributed to the decrease in the C content in the martensite. Based on the Fe–C phase diagram, the C content in martensite ($C_M$)

quenched from 750 °C is expected to be ~0.58 wt.%, while it is ~0.26 wt.% for the sample quenched at 810 °C. The strong influence of C content on the martensite strength has been also discussed in prior studies [57,100,101,126]. Notably, the carbon content in the ferrite phase is expected to exhibit only a negligible increase, which does not appear to influence significantly the indentation yield strength of the ferrite.

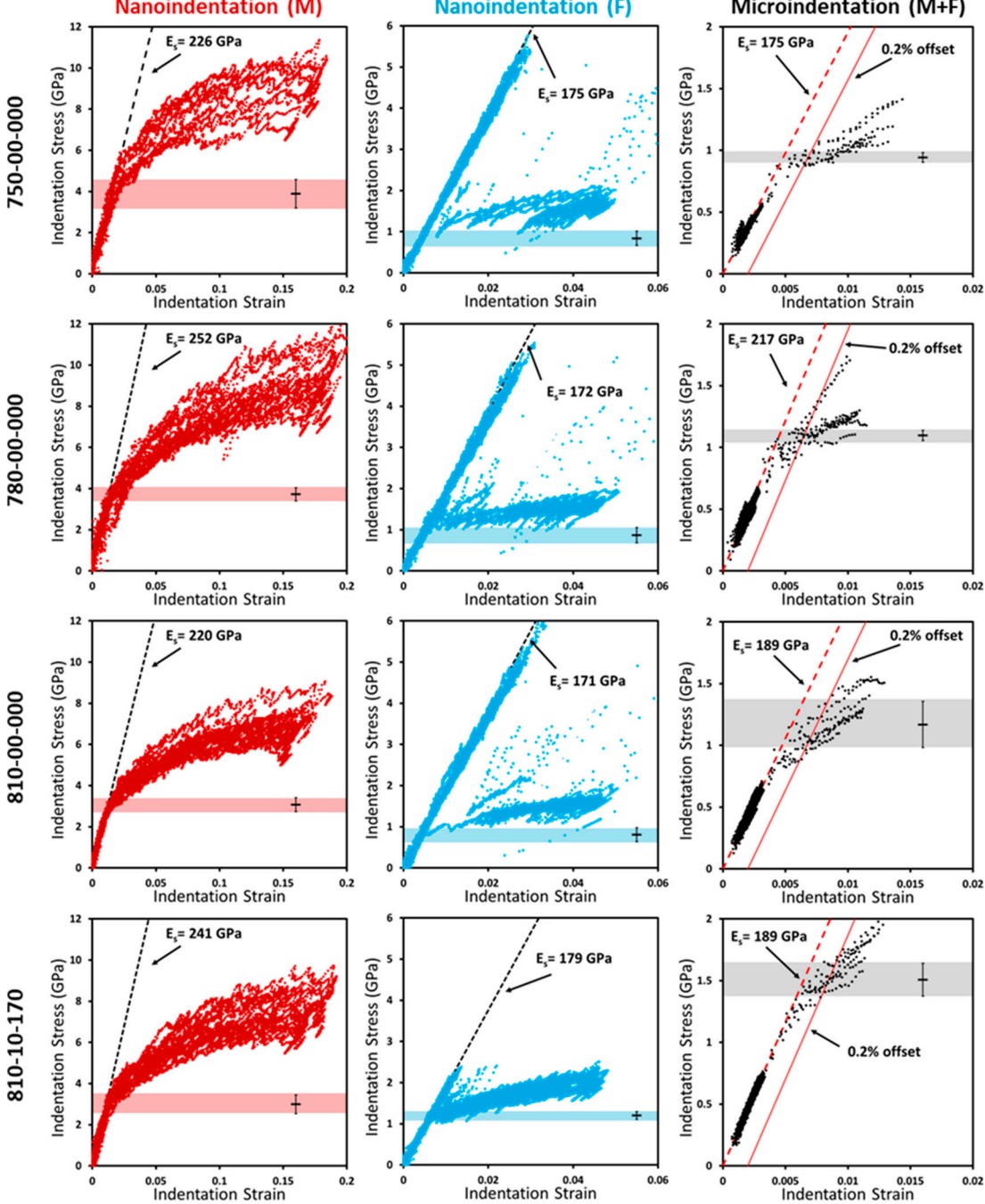

**Figure 8.** Multiple indentation stress–strain measurements at different lengths scales on samples 750-00-000, 780-00-000, 810-00-000, and 810-10-170. The left column shows the indentation stress–strain curves from multiple tests on martensite using the 1 μm tip radius. The center column shows the indentation stress–strain curves from multiple tests on ferrite grains using the 16 μm tip radius. The right column shows the micro-indentation stress–strain curves from multiple tests using the 6.35 mm tip radius. The highlighted horizontal bands inside each plot show the average values of the indentation yield strength and their variations.

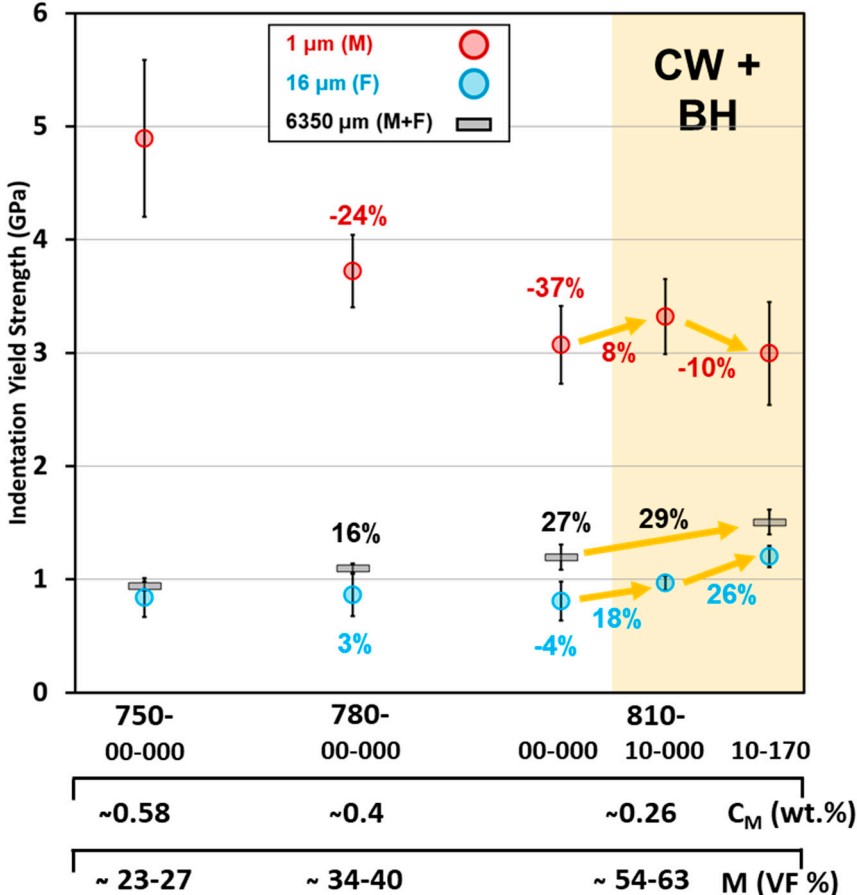

**Figure 9.** Extracted indentation yield strengths for samples 750-00-000, 780-00-000, 810-00-000, 810-10-000, and 810-10-170 at different length scales in martensite (M) and ferrite (F) regions. Bulk measurements (M + F) are obtained using microindentation tests. The numbers in the white region show the percentage changes with respect to the values from sample 750-00-000. The numbers in the yellow box show the percentage changes during the cold work and tempering steps in the bake hardening process for the 810-10-170 sample. The percentage of the martensite volume fractions and the carbon concentration in martensite are estimated from the Fe–C phase diagram in Figure 1.

Comparing the measurements on the martensite regions in sample 810-00-000 with those on sample 810-10-170 in Figure 9, it is seen that the average indentation yield strength decreased by a negligible amount (~2.5%) after the bake hardening step. This observation is consistent with other studies where it was reported that martensite hardness remains unchanged when the bake hardening temperature is kept below ~250 °C [126–130]. To separate the effects of hardening during cold work and softening by tempering in the bake hardening process, another sample was prepared right after cold work and without applying bake hardening. This sample is labeled as 810-10-000 in Figure 9. The indentation results show an increase of 8% in the indentation yield strength of the martensite regions during the cold work, followed by 10% softening during the tempering process in the bake hardening step. It was observed that during the bake hardening process, carbon atoms diffuse out of martensite islands into low carbon content ferrite grains and tend to build a carbon Cottrell atmosphere around dislocation cores, which immobilizes the dislocations [11,34,131–133]. Figure 10 shows an example of a ferrite grain surrounded by martensite islands in sample 810-10-170. As seen in Figure 10a, the ferrite grain has a highly contrasted matrix, which makes it very difficult to identify its interface with neighboring martensite islands. The ferrite grain boundaries are highlighted by dashed lines colored in yellow in this figure. Higher resolution images from the ECCI technique (see Figure 10b,c) reveal the presence of highly dense dislocation networks inside the contrasting regions. Few dislocations are highlighted by

blue arrows in Figure 10c, which appear as white contrast on the black-background matrix. As shown in the highlighted yellow area in Figure 9, the indentation yield strength of ferrite increased by 50% when cold work and bake hardening were applied (from 0.81 ± 0.17 GPa for sample 810-00-000 to 1.2 ± 0.09 for sample 810-10-170). Indeed, one should expect an increase in the yield strength of the ferrite due to the imposed cold work. However, ferrite exhibits a low work hardening rate and the applied cold work only increased the indentation yield strength by 18% in the sample 810-10-000. A further 26% increase was observed in the indentation yield strengths in the ferrite between the samples 810-10-000 and 810-10-170, which is attributed to static aging resulting from the pinning of the dislocations by the diffused carbon atoms from the martensite.

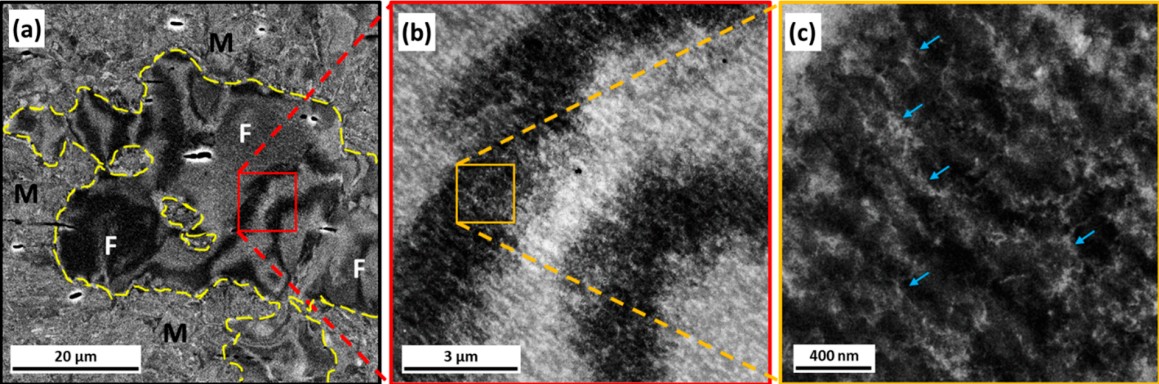

**Figure 10.** (**a**) BSE-SEM micrograph obtained on sample 810-10-170. The high contrast in both ferrite and martensite makes it difficult to identify their interface. For this purpose, each phase is labeled as martensite (M) or Ferrite (F) and their interfaces are highlighted as yellow dashed lines. (**b**) The ECCI technique at higher magnifications reveals a dense network of dislocations inside a ferrite grain. (**c**) The dislocation cores can be distinguished at sub-micron magnifications (see blue arrows).

Indirect evidence of a significant role of dislocation pinning on the strengthening of the ferrite can be found in indentation results. All the collected indentation load–displacement curves from ferrite regions on sample 810-10-170 revealed significant pop-in events while no pop-ins were observed from ferrite regions on sample 810-10-000. Figure 11a shows an example of a 3.6 nm pop-in (highlighted by a red arrow) in the indentation depth range of 30–40 nm. In prior studies using the spherical indentation protocols described earlier, pop-ins were observed exclusively on fully-annealed metal samples when small indenter tips (e.g., a spherical tip with 1 μm radius) were utilized [54,122,134–136]. In those studies, pop-ins disappeared after the introduction of small amounts of plastic deformation [136]. Therefore, in the prior studies, it was concluded that the pop-ins were a consequence of the difficulty of activating dislocation sources within the primary indentation deformation zone [54,120–122,124,125]. In the present work, the lack of pop-ins in the 810-10-000 sample is consistent with the previous observations described above. However, the 810-10-170 sample showed significant pop-ins, even though the ferrite grains have multiple dislocations within the indentation zone size of the 16 μm radius tip (see Figure 10). In the indentations performed on ferrite regions in this study, the ratio of the pop-in stress to the back-extrapolated yield strength was found to be 1.63 ± 0.35. This means that the initiation of plastic deformation was found to be significantly more difficult than continued plastic deformation after the initial yield. Therefore, the pop-ins observed in the tests reported here on the ferrite regions are attributed to the pinning of dislocations by static aging described earlier. Furthermore, as indentation proceeds, the indentation primary zone expands and encompasses new pinned dislocations. If an insufficient number of mobile dislocations exist inside the indentation primary zone, one should expect the occurrence of additional pop-ins. In our experiments, we often found such additional pop-ins (see the one marked by the orange arrow at the depth of 40 nm in Figure 11a). Of course, the subsequent pop-ins do not produce as large drops in stresses as the response is averaged over a much larger volume. It is important to note that such additional pop-ins were

rarely observed in prior studies on fully annealed metal samples [54,58,61] because the dislocation sources are easily established in these samples at the larger indentation zone sizes created after the first pop-in. The presence of these pop-ins in the 810-10-170 and their absence in 810-10-000 provides the strongest direct evidence supporting the hypothesis that the ferrite regions in the DP steels experience a significant amount of dislocation pinning during the bake hardening step.

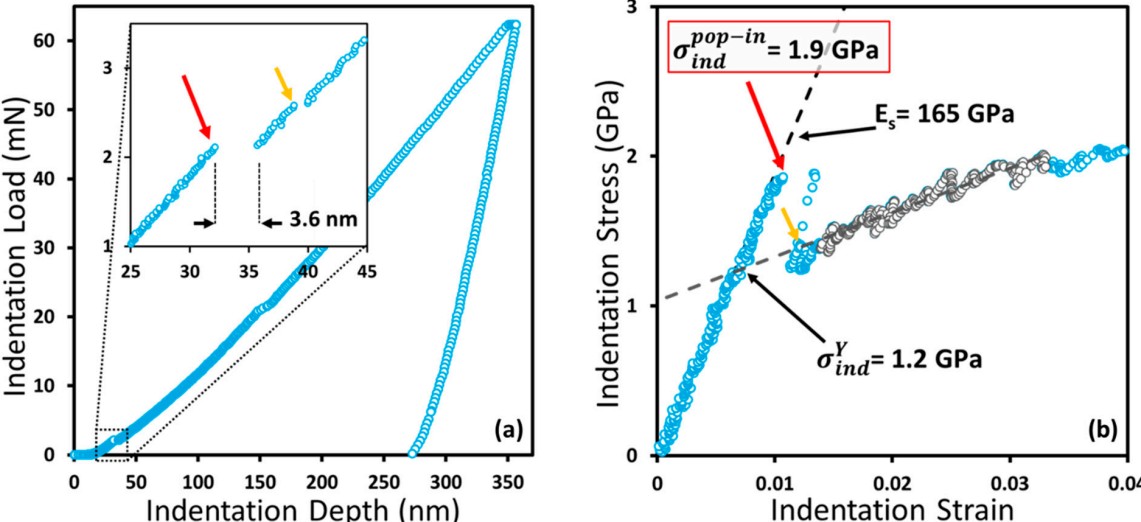

**Figure 11.** Example (**a**) load–displacement and (**b**) indentation stress–strain curves on the ferrite region from sample 810-10-170. The observed pop-in is attributed to the higher stress needed to unpin the dislocations from the carbon atmosphere.

## 4. Conclusion

The mechanical properties of DP steels were investigated at different length scales using spherical indentation stress–strain protocols. A total of nine samples processed with different combinations of intercritical annealing temperatures and different amounts of cold-work in the bake hardening step were studied. Bulk property measurements were carried out using a 6.35 mm tip radius showed results consistent with the relevant existing tensile data from the literature. However, the spherical indentation protocols used in this study require very small material volumes. An increase of the martensite volume fraction from 25% to 56% (for samples quenched from 750 °C and 810 °C, respectively) resulted in a 27% increase of the bulk indentation yield strength. Furthermore, the addition of the bake hardening step to the sample subjected to intercritical annealing at 810 °C produced a further 29% increase in the indentation yield strength.

Nanoindentation measurements were performed separately on both martensite and ferrite regions to obtain quantitative insights into how the different processing parameters affected their individual properties (at the scale of individual phases). It was first established that the measurements on martensite are best conducted using the 1 μm radius indenter tip, while the measurements on ferrite needed the 16 μm radius indenter tip for producing the most reliable data. The measurements conducted in this study revealed that the martensite regions exhibited on average a 36% higher Young's modulus compared to the ferrite regions. Intercritical annealing at 810 °C instead of at 750 °C decreased the indentation yield strength of martensite by ~37%, while the indentation yield strength of the ferrite regions remains more or less unchanged. Cold work and bake hardening of the sample intercritically annealed at 810 °C, increased the indentation yield strength in the ferrite regions by 50% without a significant change in the indentation yield strength of the martensite. The effects of cold work and tempering treatments employed in bake hardening were investigated. The results showed that the cold work increased the indentation yield strength of both phases, while the tempering treatment softened the martensite and hardened the ferrite. Most importantly, the nature of the pop-ins observed in the

nanoindentations performed on the ferrite regions suggested that the increase in the indentation yield strength during bake hardening can be attributed to the pinning of dislocations by the diffused carbon.

**Author Contributions:** A.K. and S.R.K. conceived and designed the experiments; A.K. and C.M.C. performed the experiments; A.K. analyzed the data; A.K. and S.R.K. wrote the paper. All authors have read and agreed to the published version of the manuscript.

**Funding:** The authors gratefully acknowledge support from the National Science Foundation through the funding grant 1761406.

**Conflicts of Interest:** The authors declare no conflict of interest–exists.

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
