# Peer review of "New Insights into the Microstructural Changes During the Processing of Dual-Phase Steels from Multiresolution Spherical Indentation Stress–Strain Protocols"

_metals, doi:10.3390/met10010018_

Round 1

Reviewer 1 Report

The paper presents a thorough study on the use of nano-hardness to study dual phase steels. It is very well written and the figures are very professional. I have only two comments for the authors. Firstly, the paper is quite long, and the authors might consider some sub headings to help the reader navigate such a large text. Secondly, the authors contend that the increase in ferrite hardness is due to carbon atmospheres pinning dislocations rather than work hardening. This argument would be more compelling if the authors measured the ferrite hardness after rolling but before heating. This would categorically prove their hypothesis. The ferrite also has dislocations before bake hardening, and these too are pinned, so one might content that the increase in hardness after bake hardening is due to both the increase in dislocation density plus the pinning effect. 

Author Response

Suitable sub-headings have been introduced in the revised version.

We prepared another sample according to the reviewer’s suggestion. The sample was labeled as 810-10-000, which indicated the sample were quenched from 810C and subjected to 10% cold work. Indentation measurements were collected on both ferrite and martensite regions, and the results were added and discussed in the revised text.

Reviewer 2 Report

Dear authors, 

please clarify and/or take account to increase quality of your paper:

1.- pp.2 Figure 1. It is not Fe-C equilibrium diagram. It is partial diagram Fe-C

2.-pp.4 you say " The heat treatment was carried out in a salt bath" sure? why? with is this method? with is the chemical composition of this bath? I consider that it is a mistake that you must solve.

3.- pp 13 and others, you say "As already mentioned, these samples produce martensite regions with significant differences in C content"....Please clarify the way that you do measurements of C content for each phase and evolution. with your thermal treatments. I consider that is necessary to explain Mn effect.

4. pp. 15 you say "Based on the Fe-C phase diagram, the C content in martensite quenched from 750 °C is expected to be ~0.58 at%, while it is ~0.26 at% for the sample quenched at 810 °C".please consider the scale: %wt or %at

5. it would be interesting that deep in future work the effect of interphases M/F since  can provide interesting results, in particular related to accumulation of C.

Best greetings,

Author Response

1. pp.2 Figure 1. It is not Fe-C equilibrium diagram. It is partial diagram Fe-C

Modifications were made on the figure caption and in the text

2. pp.4 you say " The heat treatment was carried out in a salt bath" sure? why? with is this method? with is the chemical composition of this bath? I consider that it is a mistake that you must solve.

It was already mentioned in the paper that salt bath was chosen in order to ensure the entire sample uniformly heated up and quickly reached the intercritical annealing temperature. The heat treating salt was a commercial salt, LIQUID HEAT 168, from Houghton International with operating temperature between 730-900 °C. The salt is suitable for heat treatment of a variety of steel alloys. This salt is suitable for steel alloys as their compositions do not react with the sample.

3. pp 13 and others, you say "As already mentioned, these samples produce martensite regions with significant differences in C content"....Please clarify the way that you do measurements of C content for each phase and evolution. with your thermal treatments. I consider that is necessary to explain Mn effect.

Exact measurement of the carbon content is very expensive process by APT technique. Which requires preparing a sample at sub-micron dimensions. Furthermore, such composition measurements cannot represent the entire sample since the local composition at the nanometer scale varies from point to another point. That is why that we only used the Fe-C phase diagram to estimate the C content in each phase. The effect of the Mn was not investigated in this study since the diffusion of Mn atoms during the bake hardening process to far distance is very negligible and considered to remain unchanged. In addition, the effect of Mn on the tensile strength of steel is very small compared to the effect of carbon. Investigating Mn effect can be a target for a future study if samples with systematic changes of Mn content become available.

4. pp. 15 you say "Based on the Fe-C phase diagram, the C content in martensite quenched from 750 °C is expected to be ~0.58 at%, while it is ~0.26 at% for the sample quenched at 810 °C".please consider the scale: %wt or %at

This is fixed in the revised version. It is now consistent in the entire text.

5. It would be interesting that deep in future work the effect of interphases M/F since can provide interesting results, in particular related to accumulation of C.

It is a very thoughtful suggestion. This study is an ongoing project and we have thought about this suggestion as the next step in our study